# The Psychophysiological Profile and Cardiac Autonomic Reactivity in Long-Term Female Yoga Practitioners: A Comparison with Runners and Sedentary Individuals

**DOI:** 10.3390/ijerph19137671

**Published:** 2022-06-23

**Authors:** Jia-Ru Lin, Pei-Tzu Wu, Wen-Lan Wu, Yu-Kai Chang, I-Hua Chu

**Affiliations:** 1Department of Sports Medicine, College of Medicine, Kaohsiung Medical University, Kaohsiung 807, Taiwan; tuyet527@gmail.com (J.-R.L.); wenlanwu@kmu.edu.tw (W.-L.W.); 2Division of Metabolism, Kaohsiung Municipal Feng-Shan Hospital, Kaohsiung 830, Taiwan; 3School of Physical Therapy & Athletic Training, Pacific University, Hillsboro, OR 97123, USA; eliza.wu@pacificu.edu; 4Ph.D. Program in Biomedical Engineering, College of Medicine, Kaohsiung Medical University, Kaohsiung 807, Taiwan; 5Department of Physical Education and Sport Sciences, National Taiwan Normal University, Taipei City 106, Taiwan; yukaichang@ntnu.edu.tw; 6Institute for Research Excellence in Learning Science, National Taiwan Normal University, Taipei City 106, Taiwan; 7Department of Medical Research, Kaohsiung Medical University Hospital, Kaohsiung 807, Taiwan

**Keywords:** yoga, running, heart rate variability, cardiovascular reactivity, stress response

## Abstract

Yoga practice, a means of stress management, has been reported to optimize psychophysiological health; however, its underlying mechanisms remain unclear. The purpose of the present study was to examine the psychophysiological profile and cardiac autonomic reactivity in long-term yoga practitioners and compare them to runners and sedentary individuals. Psychological health and aerobic fitness level were evaluated using self-reported questionnaires and a 3-min step test. Blood pressure (BP), heart rate (HR), respiration rate (RR), and heart rate variability (HRV) parameters were recorded at rest, as well as during and following psychological stress, which was elicited by the Stroop color and word test and the mental arithmetic task. The yoga group demonstrated a lower RR (10.35 ± 2.13 bpm) as compared to the other two groups, and a lower HR (66.60 ± 7.55 bpm) and diastolic BP (67.75 ± 8.38 mmHg) at rest when compared to the sedentary group (all *p* < 0.05). HRV parameters following mental stress returned to the baseline in yoga and running groups, but not in the sedentary group. The anxiety level in the running group was significantly lower than that in the sedentary group (*p* < 0.05). These findings suggested that yoga practitioners may have a greater homeostatic capacity and autonomic resilience than do sedentary individuals.

## 1. Introduction

Yoga, a mind–body practice, has become popular worldwide as a means of potentially improving physiological and psychological health [1,2]. Yoga practice has been found to reduce levels of perceived stress [2,3,4,5], anxiety [2,4,6,7] and depression [2,4,8], as well as improve quality of sleep [9,10] and quality of life [4,10]. Although growing publications have investigated the relationship between yoga practice and stress management, its underlying mechanisms remain unclear, primarily due to methodological limitations [11].

Studies have also demonstrated that yoga practice reduces cardiovascular risk, such as total cholesterol, triglycerides, blood pressure (BP), waist circumference, body mass index (BMI), and fasting glucose level, and these findings were consistent in both the population without [12] or with existing cardiovascular risk factors [12,13,14,15,16]. However, among these published studies, the dosage (from three times weekly to 3 to 4 h daily) and duration (from 8 days to 1.5 years) of intervention greatly varied. In addition, limited studies investigated the effects of long-term (>1.5 years) yoga practice on cardiovascular benefits [17,18]. 

The clinical importance of heart rate variability (HRV) was recognized in the late 1980s, when it was considered as an independent predictor of mortality after cardiovascular events [19,20,21]. It is widely accepted that HRV contributes to the understanding of the autonomic regulation of heart rate (or RR intervals) fluctuations [22]. HRV increases during relaxation and decreases during stress, and is associated with mental health (e.g., anxiety and depression) and lifestyle factors (e.g., physical activity, smoking) [7,8,23,24]. The variations in heart rate are most commonly evaluated by time-domain (e.g., standard deviation of the normal-to-normal intervals, SDNN) or frequency-domain (e.g., power in high-frequency range, HF) measures. Individuals who regularly exercise, compared to their sedentary peers, demonstrated higher frequency-domain and time-domain [25] measures of HRV. The elevated HRV presented in regular exercisers reflects increased parasympathetic nervous system activity, which partly explains the pronounced resting sinus bradycardia commonly found in endurance-trained athletes [25]. Peter et al. measured HRV in more than 1000 healthy male volunteers, comprising yoga practitioners, athletes, and sedentary individuals. A significant reduction of the low-frequency/high-frequency ratio was noted in yoga practitioners when compared to athletes and sedentary individuals, suggesting greater parasympathetic activity in yoga practitioners [26]. Yoga intervention also demonstrates effects in improving HRV. Chu et al. found that HRV in women with depressive symptoms was improved after 12 weeks of yoga training [8], and similar results were also noted in healthy women [7].

Furthermore, training status affects not only HRV at rest but also HRV’s response to stress. While a decrease of HRV was noted when people underwent mental stress stimuli [27,28], a greater decline in HRV during and following mental stress stimuli was observed in regular exercisers when compared to untrained individuals [29]. Yoga practitioners also demonstrated the greatest reactivity to stress and the most rapid recovery after stress when compared to non-yoga practitioners [30]. These findings suggest that yoga practitioners may have a greater homeostatic capacity and autonomic resilience. Satin et al. reported that yoga practitioners and runners showed favorable psychophysiological parameters of cardiovascular health when compared to sedentary individuals. However, HRV was assessed only by frequency-domain measures (i.e., HF) [31]. To comprehensively assess cardiovascular reactivity to mental stress stimuli among these populations, an analysis of HRV parameters including time-domain measures (e.g., SDNN) is warranted.

The purpose of this study was to compare the psychophysiological parameters at rest, as well as during and after stress stimuli, among long-term yoga practitioners, runners, and sedentary individuals.

## 2. Materials and Methods

### 2.1. Participants

Female individuals were recruited if they (1) were aged between 20 and 50 years old, and (2) had BMI between 18.5 and 30 kg/m^2^. The inclusion criteria were based on previously published research which revealed that sex, age (younger vs. above 50 years) and BMI (above 30 kg/m^2^) led to considerable differences in HRV indices and elevated cardiovascular risk [32,33,34]. A total of 60 participants were recruited and screened for eligibility through telephone calls or emails. Participants who presented or reported any of the following conditions were excluded from the study: (1) history or current cardiovascular disease, diabetes mellitus or hypertension; (2) neurological diseases (e.g., history or current cerebrovascular accidents, autonomic neuropathy); (3) musculoskeletal conditions that would affect aerobic fitness test; (4) substance abuse; (5) current smokers; (6) pregnant or breastfeeding women. Participants who refused to complete the electrocardiogram test (ECG) and survey questionnaires were also excluded from the study. 

To be qualified for the yoga or running group, participants had to regularly perform yoga or running for ≥30 min per session, ≥3 times a week (or for ≥90 min per week), for at least 2 years. Participants who did not regularly perform any type of physical activity were assigned to the sedentary group. The sample size for this study was calculated using G*Power version 3.1 (Heinrich Heine University, Düsseldorf, Germany) [35] with alpha of 0.05, power of 0.80 and the effect size of 0.25. The a priori analysis yielded a sample size of 18 for each group to achieve statistical power of 0.83. The study protocol was approved by the Kaohsiung Medical University Chung-Ho Memorial Hospital Institutional Review Board (KMUH-IRB-E(II)-20170246). All participants were informed of the benefits and risks of the investigation and signed a written informed consent.

### 2.2. Self-Report Measures

Perceived stress was measured by the 14-item Perceived Stress Scale (PSS-14). Each item was rated on a 5-point Likert scale, ranging from 0 to 4. The total score was calculated by summing the points across all the 14 items, ranging from 0 to 56. A higher score indicated a higher level of perceived stress. This scale has demonstrated adequate internal consistency, with Cronbach’s alphas ranging from 0.84 to 0.86.

Depressive symptoms were measured using the Beck Depression Inventory-II (BDI-II), a self-report inventory with 21 items assessing the behavioral and cognitive symptoms of depression. Each item was rated on a 4-point scale ranging from 0 to 3. The total score was calculated by summing the ratings for each of the 21 items, ranging from 0 to 63, with scores 0–13 indicating minimal depression, 14–19 indicating mild depression, 20–28 indicating moderate depression, and 29–63 indicating severe depression. This inventory has demonstrated good internal consistency, with Cronbach’s alphas ranging from 0.92 to 0.93. 

Anxiety levels were assessed using the State-Trait Anxiety Inventory (STAI). STAI has 40 items in total, including 20 items allocated to each of the state anxiety (SA) and trait anxiety (TA) subscale. All items were rated on a 4-point Likert scale, ranging from 1 to 4. Item scores were added to obtain a total score for each subscale. The higher score indicated greater anxiety. The STAI has shown good internal consistency, with Cronbach’s alphas ranging from 0.86 to 0.95.

Pittsburgh Sleep Quality Index (PSQI) [36], a self-rated questionnaire, was used to assess sleep quality over a 1-month period. The scores of each PSQI category were rated from 0 to 3, with a total score ranging from 0–21 by summing the scores from all 7 categories. The PSQI has good internal consistency (Cronbach’s alpha 0.83) and test–retest reliability [36].

The Godin Leisure-time Exercise Questionnaire (GLEQ) [37,38] was used to measure weekly physical activity level. Each participant reported the number of times he/she engages in mild (e.g., yoga), moderate (e.g., brisk walking) and strenuous (e.g., running) leisure-time physical activity bouts of at least 15 min in duration in a typical week. The score of weekly leisure activity was calculated by the following formula: Weekly leisure activity = (9 × number of bouts at strenuous) + (5 × number of bouts at moderate) + (3 × number of bouts at mild).

### 2.3. Physiological Measures

All physiological measures were conducted in the laboratory by two trained experimenters. Heart rate and respiration rate (RR) were recorded using an ECG system (ProComp Infiniti™System, T7500M, Thought Technology Ltd., Montreal, QC, Canada). The electrodes were placed at mid-clavicular points bilaterally and with a third electrode placed 1 cm above the left lower costal margin. The respiratory sensor was connected with a loop rubber strap that was placed around the participant’s abdomen, approximately 2 cm above the umbilicus. 

Heart rate variability parameters, including standard deviation of normal-to-normal (SDNN), the square root of the mean squared differences of successive normal-to-normal intervals (RMSSD), total power (TP), and high frequency normalized unit (HFnu), were analyzed using the Chart 5 for Windows (ADinstruments, Bella Vista, Australia). HRV was analyzed at rest, during, and 5-min after the mental stress stimuli (i.e., SCWT and MAT).

Three-minute step test was used to evaluate participants’ aerobic fitness level. Participants were instructed to step up and down on a 35-cm box for 3 min. Stepping frequency was set at 96 beats per minute (4 movements = one step cycle) for a stepping rate of 24 steps per minute, of which a fully calibrated audio file was downloaded from the Sports Administration, Ministry of Education website [39]. The 3-min audio file was played from the beginning to the end of the step test for each participant. The participants immediately stopped upon completion of the test and then were instructed to sit down and remain still. The participant’s heart rate (HR) was monitored and recorded at the 90th, 150th, and 210th second during the recovery phase (Polar Electro Oy, FIN-90440 KEMPELE, Finland). The fitness index was calculated using an equation formulated by the Sports Administration, Ministry of Education [40] as follows:(1)$$Aerobic\;fitness\;index=\frac{Exercise\;duration\;\left(seconds\right)\times 100}{\left(Sum\;of\;the\;3\;HRs\;measured\;during\;the\;recovery\;phase\right)\times 2}$$

### 2.4. Stressors

The Stroop Color and Word Task (SCWT) [41] was used to induce mental stress. During the task, color-words were presented to the participants via Microsoft PowerPoint slides, in an incongruent color ink (for instance the word “red” is printed in green ink). Participants were required to name the color of the ink instead of reading the word. The PowerPoint slide was presented to the participant at a speed of 1 word per second for a period of 5 min. 

The Mental Arithmetic Task (MAT) was performed for 5 min following the Stroop task. Participants were asked to mentally perform serial subtraction tasks and respond as quickly and accurately as possible. An experimenter administrating the task would say “error” and provide the participant the correct number if the participant gave an incorrect answer. The subtrahend was reset each minute to maintain a high level of task difficulty and involvement. 

### 2.5. Procedures

All participants were requested to visit the laboratory between 8 am and 11 am. They were instructed to avoid strenuous exercise 24 h prior to the test, and avoid food or drink containing alcohol or caffeine 4 h prior to the test. After signing the informed consent form, participants completed the PSS-14, BDI-II, STAI, PSQI, and GLEQ in order. Next, participants’ height and weight were measured without shoes using a portable stadiometer (H-101, Ming-I Medical Instrument, Kaohsiung, Taiwan) and a digital weighing scale (OMRON SC-150, OMRON Healthcare Co., Ltd., Kyoto, Japan).

After 15 min of rest in supine position, the participants’ BP was measured (OMRON HEM-7200, OMRON Healthcare Co., Ltd., Kyoto, Japan), followed by the measurements of 5 min of ECG. Participants were then instructed to perform 5 min of the SCWT and 5 min of the MAT, with continuous BP and ECG monitoring throughout the mental stress phase as well as during the 5-min recovery phase. After the recovery phase, participants were provided an additional rest period of 10 min to allow HR to return to resting value. Next, the experimenter demonstrated the step test procedures and instructed the participants to perform a 5-min warm-up. Participants then performed the 3-min step test and concluded the study (Figure 1).

### 2.6. Statistical Analysis 

All data are presented as mean ± standard deviation unless stated otherwise. One-way ANOVA and Bonferroni post-hoc comparison was used to compare the differences of variables among the 3 groups (i.e., yoga, running, and sedentary). Heart rate and BP in the 3 groups were compared at 5 phases (i.e., baseline, SCWT, MAT, the 1st and 5th minute recovery phases) using a 3 × 5 (Group × Time) mixed design analysis of variance (ANOVA). Parameters of HRV in the 3 groups were compared at 4 phases (i.e., baseline, SCWT, MAT, and the 5th minute recovery phase) using a 3 × 4 (Group × Time) mixed design ANOVA. The Greenhouse–Geisser correction was applied where appropriate. The within-group comparisons between different phases and baseline values were performed using paired t tests. Partial eta squared (η^2^) was reported to present the effect size. Statistical analyses were performed using IBM SPSS Statistics ver. 20.0 (IBM Co., Armonk, NY, USA). Significance level was set at *p* < 0.05.

## 3. Results

A total of 57 participants completed the study, including 20 in the yoga group, 19 in the running group, and 18 in the sedentary group.

The characteristics of the participants are displayed in Table 1. BMI was significantly higher in the sedentary group when compared to the yoga group. RR in the yoga group was significantly lower than that in the other two groups. While resting HR in the running group was the lowest, resting HR in the sedentary group was the highest among the three groups. Diastolic BP in the sedentary group was significantly higher than that in the yoga group. The GLEQ score in the running group was significantly higher than that in the other two groups. The yoga group also showed a significantly higher GLEQ score than the sedentary group. The aerobic fitness index was the highest in the running group when compared to the yoga and sedentary groups. The yoga group also showed a significantly higher fitness index than the sedentary group. According to the national norms for the aerobic fitness index [42], both the yoga and running groups are categorized as “very good” (highest level), while the sedentary group is categorized as “average”.

The average years of practice (i.e., yoga for the yoga group and running for the running group) were similar between the yoga and running groups. No significant difference was observed in weekly frequency, duration, and weekly duration of practice between the yoga and the running groups. The yoga practitioners reported practicing a variety of yoga styles, including Hatha (35%), Iyengar (25%), Ananda Marga (20%), Aerial (15%), and Ashtanga (5%).

Changes in the HR across the five phases were not significantly different among the three groups (F = 0.710, *p* = 0.624, η^2^ = 0.026; Figure 2). HR was significantly lower in the running group than the other two groups at each phase (all *p* < 0.05). A similar trend was observed in the yoga group when compared to the sedentary group (all *p* < 0.05), except during the first minute of the recovery phase. Within each group, the HR significantly increased from the baseline during the SCWT and MAT (all *p* < 0.01).

Changes in the systolic (F = 0.653, *p* = 0.660, η^2^ = 0.024; Figure 3a) and diastolic BP (F = 0.646, *p* = 0.688, η^2^ = 0.023; Figure 3b) across all the phases were not significantly different among groups. Diastolic BP at baseline was significantly higher in the sedentary group (75.44 ± 7.03 mmHg) than that in the yoga group (67.75 ± 8.38 mmHg; *p* = 0.007). When being compared within each group, both the systolic and diastolic BP significantly increased from the baseline to the SCWT and MAT phases (all *p* < 0.05). Diastolic BP in the sedentary group was significantly lower at the fifth minute of the recovery phase (71.44 ± 8.50 mmHg; *p* = 0.003) than at the baseline.

The changes of HFnu across the four phases were similar among the three groups (F = 1.653, *p* = 0.126, η^2^ = 0.058; Figure 4). HFnu was significantly higher in the running group (37.45 ± 18.76) than the other two groups (24.59 ± 9.91 and 20.86 ± 6.08) during the SCWT (both *p* < 0.01). When compared to baseline, HFnu was significantly lower during the SCWT and MAT within each group (all *p* < 0.01). However, during the recovery phase, there was no significant difference from the baseline in the running and yoga groups (44.73 ± 17.13 and 35.68 ± 16.01), while HFnu remained significantly lower in the sedentary group (29.94 ± 21.97; *p* = 0.005). 

The changes in the TP at each phase were significantly different among the three groups (F = 4.944, *p* < 0.001, η^2^ = 0.155; Figure 5). Total power at each phase was significantly greater in the running group when compared to that of the other two groups (all *p* < 0.01). In the sedentary group, compared to the baseline (1672.76 ± 1368.42), TP was significantly higher at the recovery (2938.33 ± 1660.10; *p* < 0.001) phase. However, compared to the baseline, no change was observed at any phase in the yoga group.

The changes in the SDNN at each phase were significantly different among the three groups (F = 3.251, *p* = 0.003, η^2^ = 0.107; Figure 6). The SDNN at each phase was significantly greater in the running group when compared to that of the other two groups (all *p* < 0.01). Within each group, the SDNN was significantly higher at the recovery phase (58.08 ± 18.98, 85.96 ± 31.11, 59.70 ± 15.20, respectively, for yoga, running and sedentary groups), compared to the baseline (all *p* < 0.05).

The changes in the RMSSD at each phase were significantly different among the three groups (F = 3.677, *p* = 0.004, η^2^ = 0.120; Figure 7). The RMSSD at each phase was significantly greater in the running group when compared to that of the other two groups (all *p* < 0.01). In the running group, the RMSSD was significantly lower at the MAT phase (53.74 ± 37.12; *p* = 0.038), compared to that at the baseline. In the sedentary group, RMSSSD was significantly higher at the recovery phase (36.03 ± 17.89; *p* = 0.017), compared to that at the baseline. However, compared to the baseline, no change was observed at any phase in the yoga group.

The results of psychological measures are presented in Table 2. Scores of the STAI, including both SA and TA, were the highest in the sedentary group among the three groups, and significantly higher than that in the running group. No significant differences among groups were observed in other psychological measures.

## 4. Discussion

The purpose of this study was to investigate the physiological and psychological profile in long-term yoga practitioners. In addition, we compared the physiological and psychological profiles and cardiovascular responses to stress in yoga practitioners to runners and sedentary individuals.

When compared to sedentary individuals, both yoga practitioners and runners showed a lower resting HR and higher aerobic fitness levels. Yoga practitioners also demonstrated a lower resting RR, lower BMI, and lower resting diastolic BP when compared to sedentary individuals. Runners showed higher TP, SDNN, and RMSSD across all phases and lower levels of anxiety when compared to sedentary individuals. When comparing the change of HRV between the baseline and the recovery phase, HF, TP, and RMSSD returned to the baseline in yoga practitioners and runners, but not in sedentary individuals. 

Yoga practice has been known for involving breathing techniques, which include a slow deep breath inspired by the use of the abdominal muscle group and diaphragm. Slow breathing has been demonstrated to enhance HRV by re-synchronizing inherent cardiovascular rhythms [1,43]. The beneficial effect has been observed not only in the RR but also in the resting HR and BP [44]. In addition, yoga practitioners demonstrated a better lung function than their athletic and sedentary peers [45]. These previous studies indicated that long-term yoga practitioners might be able to breathe more efficiently with a lower RR when compared to the runners and sedentary individuals.

The findings of lower resting HR and BP observed in yoga practitioners and runners were consistent with previous studies. Down-regulation of resting HR has been reported in individuals who participate in regular exercise or leisure-time activities The lower resting HR may be as a result of increased parasympathetic activity, improvement in central regulation of the autonomic outflow, increased left ventricular end-diastolic volume, increased left ventricular mass, or increased stroke volume [46,47]. In addition, regular yoga practice [48,49] and aerobic exercise [50] has been reported to effectively lower BP in the population with elevated BP.

All of the resting HRV parameters (i.e., HFnu, TP, SDNN, RMSSD) in runners were significantly higher than the other two groups, suggesting that long-term regular aerobic exercise was effective in increasing parasympathetic nervous activity and overall HRV. These results were consistent with findings from previous reviews and meta-analyses indicating the beneficial effects of aerobic exercise training on HRV in healthy adults [51,52]. On the other hand, the resting HRV in yoga practitioners was not different from the sedentary individuals. Few studies have examined the resting HRV in long-term yoga practitioners. Results from two cross-sectional studies showed significantly higher HFnu and RMSSD in yoga practitioners when compared to sedentary individuals [30,31]. The yoga practitioners in these two studies appeared to practice yoga more often (averaged five to seven times per week) and for a longer period of time (>6 years) as compared to our participants (averaged four times per week for 4 years). It seems that more frequent and longer yoga practice may be needed to elicit positive HRV adaptations. Further studies are needed to investigate the potential dose–response relationship between long-term yoga practice and HRV.

Greater cardiovascular responses to acute mental stress and slower stress recovery has been used to predict risk of cardiovascular disease [53]. In this study, however, we did not observe any difference in HR and BP responses to stress stimuli between the exercise groups and the sedentary group. Only two studies have examined and compared HR responses to stress between yoga practitioners and non-yoga practitioners. While one study showed significantly lower HR in yoga practitioners [54], the other study found no significant difference in HR responses between the two groups [30]. In addition, the current study is the first to report BP responses to stress in yoga practitioners. While systolic BP and diastolic BP increased significantly during the SCWT and MAT phases, both BP changes were not different among the groups. With such few studies, the effect of long-term yoga practice on HR and BP responses to stress remains inconclusive and warrants further investigation. 

In the study, compared to the baseline, the HFnu reduced during the stress tests in all three groups; however, during the recovery phase, the HFnu returned to baseline in the yoga and running groups while it remained lower than the baseline in the sedentary group. This phenomenon was similar to the findings reported by Tyagi et al. [30]. The HFnu decreased under stress in yoga practitioners and non-yoga practitioners, but HFnu in non-yoga practitioners remained at a significantly lower level during the recovery phase. These findings suggested that regular exercisers may have a greater ability to restore parasympathetic nervous activity more quickly following mental stress stimuli. Tyagi et al. also reported similar patterns in the RMSSD response to stress; however, the current study found no significant difference in RMSSD response between yoga practitioners and sedentary individuals. Although there was no difference in HF, TP, SDNN, or RMSSD between yoga practitioners and sedentary individuals during the fifth minute recovery phase in this study, the change of HF, TP, and RMSSD between baseline and the recovery phase was different between yoga practitioners and sedentary individuals. This finding indicated that, compared to sedentary individuals, yoga practitioners responded to mental stress with a faster HRV recovery, and the response was similar to the runners. So far only two studies (including the current study) have assessed HRV changes during mental stress in yoga practitioners, and to the best of our knowledge, the current study is the first study that compared the HRV recovery following mental stress stimuli between yoga practitioners, runners, and sedentary individuals. Thus, more research is needed to further examine the effect of long-term yoga practice on cardiac autonomic reactivity. 

Levels of anxiety were lower in the two exercise groups as compared to the sedentary group in this study. This finding is supported by previous studies and meta-analyses indicating that regular exercise is strongly associated with reduced anxiety, and has positive impacts on the pathophysiological processes of anxiety [55,56,57,58]. On the other hand, while numerous studies have reported the beneficial effects of regular exercise in reducing perceived stress and depressive symptoms, and in improving sleep quality, no significant difference in these psychological measures was observed between the exercise groups and the sedentary group. This could be due to the fact that we have recruited relatively healthier sedentary individuals as they reported low levels of perceived stress, depressive symptoms, and sleep problems. This potential floor effect may have prevented differences from being detected.

With a cross-sectional design, the present study intended to compare the outcome variables in yoga practitioners to runners and sedentary individuals, instead of indicating any causal relationship between yoga and psychophysiological profiles. In addition, as we recruited only female participants, the results may not be generalized to male yoga practitioners and runners. The modest changes in HR and HRV induced by our mental stressors and the relatively healthier sedentary group may have attenuated the ability of this study to find group differences in cardiac autonomic reactivity and psychological measures. The inclusion criteria for the yoga and running groups was at least 90 min of weekly practice duration with no upper limit for duration of weekly practice; therefore, the effect of different amounts of weekly physical activity (e.g., high, moderate, or low) was not accounted for in this present study. In addition, the yoga group practiced a variety of yoga styles. As the intensity of practice differs among different styles of yoga, this may have served as a confounding factor in the present study. Future research that compares the autonomic reactivity between different types of yoga practice (e.g., Hatha vs. Ashtanga style) is warranted. Due to the limited time and financial support to the present study, we aimed to achieve an acceptable statistical power (0.83) with 18 participants per group, instead of targeting a higher statistical power (e.g., 0.90 or higher) with more participants. Future research that includes both genders and individuals at risk (e.g., individuals with metabolic syndrome) is warranted to further examine the effect of long-term yoga practice on psychophysiological profiles and cardiac autonomic reactivity to stress. 

## 5. Conclusions

The present study demonstrated that long-term yoga practitioners and runners differ from sedentary individuals in various physiological and psychological parameters. The exercise-induced cardiovascular benefits may be attributed to a greater HRV at rest in the runners and a lower physiological status (e.g., RR, HR, and BP) at rest in the yoga practitioners. Both exercise groups restored parasympathetic nervous activity more rapidly following mental stress stimuli when compared to their sedentary counterparts, suggesting that regular exercisers may have a greater homeostatic capacity and autonomic resilience.

## Figures and Tables

**Figure 1 ijerph-19-07671-f001:**
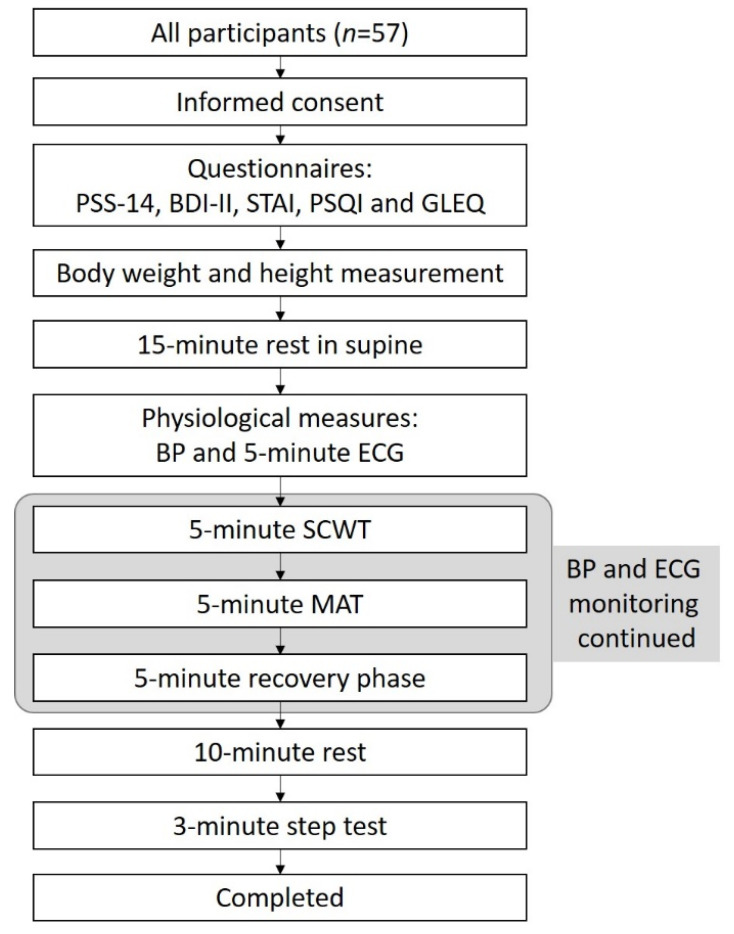
Flowchart of study procedures. PSS-14, Perceived Stress Scale-14; BDI-II, Beck Depression Inventory; STAI, State-Trait Anxiety Inventory; PSQI, Pittsburgh Sleep Quality Index; GLEQ, Goldin Leisure-time Exercise Questionnaire; BP, blood pressure; ECG, electrocardiogram; SCWT, the Stroop color and word test; MAT, the mental arithmetic task.

**Figure 2 ijerph-19-07671-f002:**
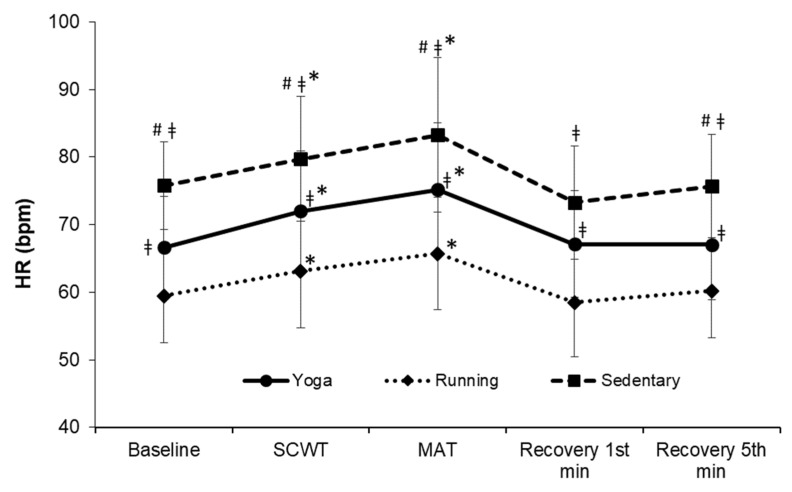
Changes in the HR over time. HR, heart rate; SCWT, The Stroop color and word test; MAT, The mental arithmetic task. ^#^ significantly different from yoga group, *p* < 0.05. ^ǂ^ significantly different from running group, *p* < 0.05. * significant difference between baseline and each phase within a group, *p* < 0.05.

**Figure 3 ijerph-19-07671-f003:**
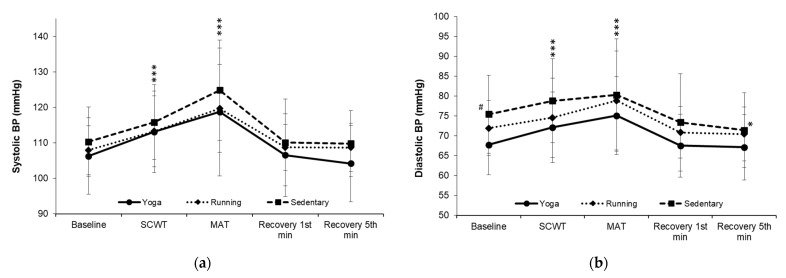
(**a**) Changes in the systolic BP over time. (**b**) Changes in the diastolic BP over time. BP, blood pressure; SCWT, The Stroop color and word test; MAT, The mental arithmetic task. ^#^ significantly different from yoga group, *p* < 0.05. * significant difference between baseline and each phase within a group, *p* < 0.05.

**Figure 4 ijerph-19-07671-f004:**
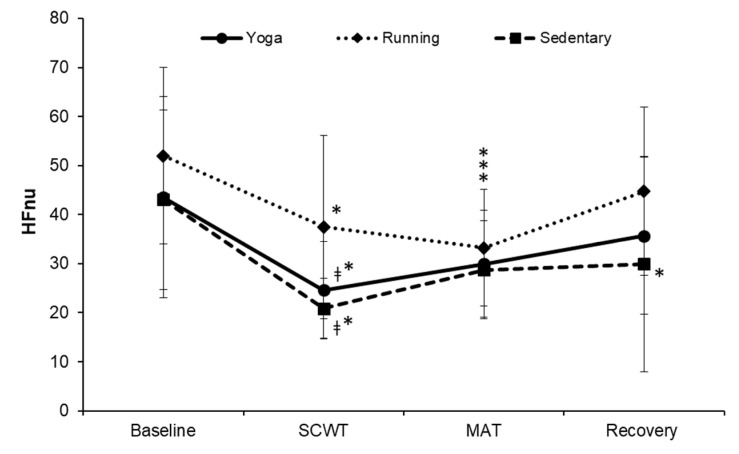
Changes in the HFnu over time. HFnu, high frequency normalize unit; SCWT, The Stroop color and word test; MAT, The mental arithmetic task. ^ǂ^ significantly different from running group, *p* < 0.05. * significant difference between baseline and each phase within a group, *p* < 0.05.

**Figure 5 ijerph-19-07671-f005:**
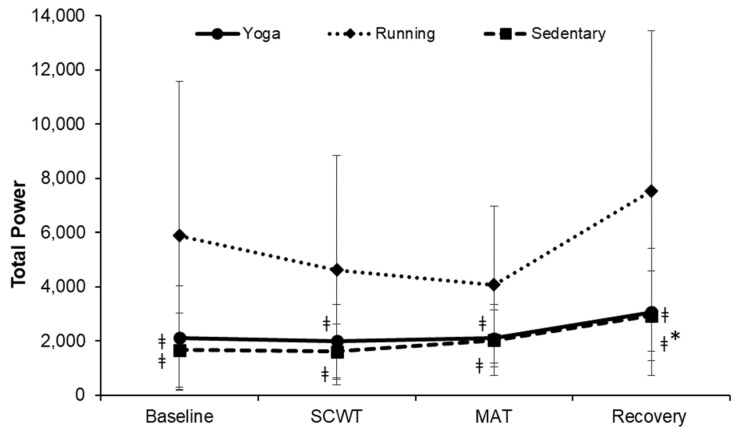
Changes in the total power over time. SCWT, The Stroop color and word test; MAT, The mental arithmetic task. ^ǂ^ significantly different from running group, *p* < 0.05. * significant difference between baseline and each phase within a group, *p* < 0.05.

**Figure 6 ijerph-19-07671-f006:**
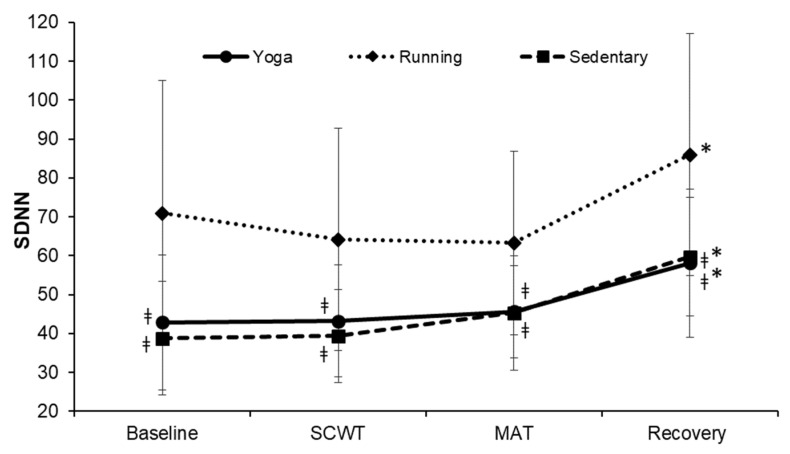
Changes in the SDNN over time. SDNN, standard deviation of normal-to-normal intervals; SCWT, The Stroop color and word test; MAT, The mental arithmetic task. ^ǂ^ significantly different from running group, *p* < 0.05. * significant difference between baseline and the recovery phase within a group, *p* < 0.05.

**Figure 7 ijerph-19-07671-f007:**
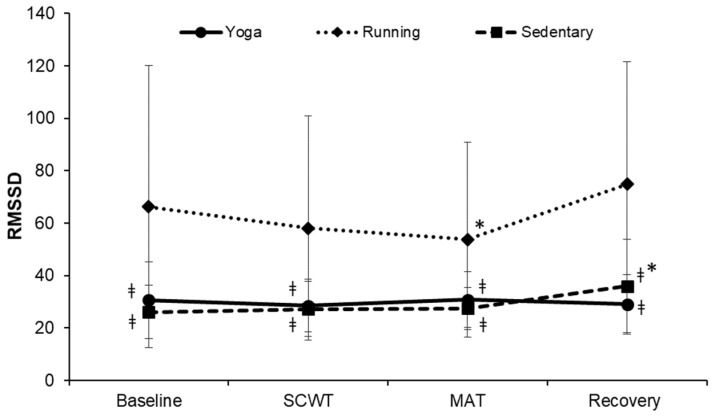
Changes in the RMSSD over time. RMSSD, the square root of the mean squared differences of successive normal-to-normal intervals; SCWT, The Stroop color and word test; MAT, The mental arithmetic task. ^ǂ^ significantly different from running group, *p* < 0.05. * significant difference between baseline and each phase within a group, *p* < 0.05.

**Table 1 ijerph-19-07671-t001:** Characteristics of the participants.

Characteristics	Yoga (*n* = 20)	Running (*n* = 19)	Sedentary (*n* = 18)	F	*p*-Value
Age (years)	37.95 ± 7.65	37.53 ± 6.58	37.33 ± 5.58	0.04	0.958
Height (cm)	161.48 ± 4.87	160.18 ± 2.56	158.17 ± 3.84 ^#^	3.45	0.039
Weight (kg)	54.35 ± 5.02	56.71 ± 5.10	58.59 ± 7.49	2.45	0.096
BMI (kg/m^2^)	20.87 ± 2.02	22.09 ± 1.81	23.43 ± 2.93 ^#^	5.95	0.005
Resting RR (times/min)	10.35 ± 2.13	13.47 ± 2.84 ^#^	14.17 ± 4.19 ^#^	8.15	0.001
Resting HR (bpm)	66.60 ± 7.55 ^ǂ^	59.47 ± 6.92	75.78 ± 6.52 ^#^^ǂ^	24.96	<0.001
Resting systolic BP (mmHg)	106.30 ± 10.79	107.95 ± 6.89	110.33 ± 9.80	0.89	0.416
Resting diastolic BP (mmHg)	67.75 ± 8.38	71.95 ± 6.58	75.44 ± 7.03 ^#^	5.16	0.009
GLEQ score	17.45 ± 9.57 ^ǂ^	36.79 ± 9.41	2.22 ± 4.65 ^#,^^ǂ^	81.17	<0.001
Years of practice (years)	4.60 ± 2.84	4.47 ± 3.32	--	0.02	0.899
Frequency of practice (times/week)	4.15 ± 2.70	3.53 ± 1.07	--	0.88	0.354
Duration of practice (min/session)	70.00 ± 25.13	61.58 ± 33.50	--	0.79	0.379
Weekly duration of practice (min/week)	273.50 ± 158.39	213.95 ± 140.98	--	1.53	0.224
Aerobic fitness index	65.59 ± 5.41 ^ǂ^	83.34 ± 15.66	52.73 ± 6.09 ^#^^ǂ^	42.26	<0.001

BMI, body mass index; RR, respiratory rate; HR, heart rate; BP, blood pressure; GLEQ, Godin lei-sure-time exercise questionnaire; LTPA, leisure-time physical activity. ^#^ significantly different from yoga group, *p* < 0.05; ^ǂ^ significantly different from running group, *p* < 0.05.

**Table 2 ijerph-19-07671-t002:** Psychological measures.

Variables	Yoga	Running	Sedentary	F	*p*-Value
	(*n* = 20)	(*n* = 19)	(*n* = 18)		
PSS-14	17.40 ± 6.12	19.11 ± 5.75	21.89 ± 7.10	2.41	0.099
State-anxiety	34.35 ± 7.88	33.00 ± 6.75	39.83 ± 6.87 ^ǂ^	4.66	0.014
Trait-anxiety	37.05 ± 7.73	36.11 ± 7.48	42.44 ± 8.05 ^ǂ^	3.59	0.034
BDI-II	3.40 ± 3.02	2.89 ± 3.67	5.56 ± 3.79	3.01	0.058
PSQI	5.25 ± 2.65	4.58 ± 2.93	6.00 ± 1.94	1.43	0.248

PSS-14, the perceived stress scale-14; BDI-II, Beck depression inventory, 2nd edition; PSQI, Pittsburgh sleep quality index. ^ǂ^ significantly different from running group, *p* < 0.05.

## Data Availability

Data are available from the corresponding author upon reasonable request.

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
