# Peer review of "The Psychophysiological Profile and Cardiac Autonomic Reactivity in Long-Term Female Yoga Practitioners: A Comparison with Runners and Sedentary Individuals"

_ijerph, 2022, doi:10.3390/ijerph19137671_

Round 1

Reviewer 1 Report

I think this is a well-designed and conducted study, reporting evidence about the association between psychophysiological profile and cardiac autonomic reactivity in long-term yoga practitioners. 

Therefore, I would recommend publication in case the authors agree to address the following issues:   

1) Materials and Methods

-      Line 86, If you pointed out in the introduction of the manuscript that from the existing literature, yoga, may have favorable psychophysiological effects of cardiovascular health why did you exclude from the sample those who had cardiovascular problems, diabetes and hypertension? is inconsistent. 

-        Line 120, Please, add the range of Cronbach's alpha value.

-  Line 125, Indicate a literature reference of the formula.

-  Line 147, Indicate a literature reference of the formula.

2) Results

-    Table 1, write in the notes what the asterisks indicate.

Author Response

1) Materials and Methods

- Line 86, If you pointed out in the introduction of the manuscript that from the existing literature, yoga, may have favorable psychophysiological effects of cardiovascular health why did you exclude from the sample those who had cardiovascular problems, diabetes and hypertension? is inconsistent. 

Response: We thank the reviewer for the comments on the benefits of yoga practice on cardiovascular health. While yoga has been shown to benefit cardiovascular health, due to the purpose of this study, we decided to focus on the general psychophysiological parameters derived from the HRV and to investigate the comparison of responses between groups. Any history or current cardiovascular disease, diabetes and hypertension may become factors influencing the results from HRV and may mask the findings between groups. Therefore, although yoga has been proved to provide benefits in patients with cardiovascular diseases, we excluded this population in order to meet the purpose of this present study. 

- Line 120, Please, add the range of Cronbach's alpha value.

Response: Thank you for the comment. The Cronbach’s alpha value has been added in the text and reference has also been provided (line 127).

- Line 125, Indicate a literature reference of the formula.

Response: Thank you. We have added references (line 129) to support the source of the formula.

- Line 147, Indicate a literature reference of the formula.

Response: Thank you. We have added reference (line 159) to support the source of the formula.

2) Results

- Table 1, write in the notes what the asterisks indicate.

Response: We have checked Table 1. The symbols that were used in Table 1 (i.e., # and Ç‚) were noted. Please advise if further explanation is required.

Reviewer 2 Report

Dear Authors

You have written an interesting study where you have examined the psychophysiological profile and cardiac autonomic reactivity in long term yoga practitioners and compared that to runners and sedentary individuals.

However, some parts need to be addressed for greater clarity of your manuscript.

Keywords should not be from the title - like yoga delete and/or replace

Introduction

Line 44-45 / you wrote ''limited studies'' this implies that there is a lack of research - not none - so only a few, so which are they? Referencing is needed, or try rephrasing the sentence.

Materials and methods

Please elaborate on your inclusion criteria - why the limit of 50 years and why the upper limit of BMI 30. Back up this with references.

How did you determine your sample size (G*Power or any other method) and power? Report

Line 84 - 60 participants/report number of males and females.

What kind of yoga were they practising? Report

Self-reported measures - report where did the participants fill these questionnaires and what was the order (as presented or random).

Paragraph 121-127 needs referencing for the GLEQ questionnaire.

In table 1, in the running group from the SD of 147 min, it is seen that some of the participants did not meet the criteria of 90 min of running per week? Please elaborate on this as this has a significant effect on your results.

How were height and weight measured? Report

Physiological measures - where and at what time were they done? Who conducted them? Report

A flowchart of procedures would be beneficial. Please add

Step test - was there any demonstration, any warm-up, report the metronome model. How long after the physiological measurements was this done? Report

Stressors - what results were recorded and put forward into analysis and where are the descriptive results presented as they are not in table 1? report

How were your participants divided into 3 groups? report

The limitations of the study paragraph could be extended.

Overall the paper still needs a comprehensive update from the authors to increase its clarity. Therefore, I recommend major revision.

Author Response

Keywords should not be from the title - like yoga delete and/or replace

Response: Thank you for the comment. We believe that the current keywords are specific and relevant to the main topic of this current study which capture the essence of our manuscript. In addition, current keywords do not seem to violate the IJERPH requirements: “Keywords: Three to ten pertinent keywords need to be added after the abstract. We recommend that the keywords are specific to the article, yet reasonably common within the subject discipline.” (https://www.mdpi.com/journal/ijerph/instructions#front). However, we welcome any suggestions for keywords.

Introduction
Line 44-45 / you wrote ''limited studies'' this implies that there is a lack of research - not none - so only a few, so which are they? Referencing is needed, or try rephrasing the sentence.

Response: Thank you for the comment. We have added references to the manuscript (line 45) to reflect current studies on cardiovascular health in long-term yoga practitioners.

Materials and methods

Please elaborate on your inclusion criteria - why the limit of 50 years and why the upper limit of BMI 30. Back up this with references.

Response: Thanks for the comment. While we tried our best to include participants from a wide range of diversity, current inclusion criteria were selected in order to meet the purpose of this present study by minimizing the influence of sex, age and BMI on outcome variables. Relevant evidence has been added to the manuscript (line 84).  

How did you determine your sample size (G*Power or any other method) and power? Report

Response: Thank you for the comment. The sample size was calculated using G*Power version 3.1 with alpha of 0.05 and power of 0.80. The a priori analysis yielded a sample size of 18 for each group. We also added the sample size calculation to the manuscript (line 98).

Line 84 - 60 participants/report number of males and females.

Response: This study was conducted in female yoga practitioners, as indicated on the title. We also revised the methods to clarify this confusion (line 83).

What kind of yoga were they practising? Report

Response: The yoga practitioners reported practicing Hatha (35%), Iyengar (25%), Ananda Marga (20%), Aerial (15%), and Ashtanga (5%). We have added this information in the text (line 232)

Self-reported measures - report where did the participants fill these questionnaires and what was the order (as presented or random).

Response: Participants filled out the questionnaires in the laboratory in the presented order. We have added this information in the text (line 177)

Paragraph 121-127 needs referencing for the GLEQ questionnaire.

Response: Thank you. We have added references to the manuscript (line 129).

In table 1, in the running group from the SD of 147 min, it is seen that some of the participants did not meet the criteria of 90 min of running per week? Please elaborate on this as this has a significant effect on your results.

Response: Thank you for the comment. We checked the entire data and found one error in numeric value in the running group. We have rerun the analysis and corrected the data in Table 1. Besides this error, we confirmed that all participants in the running and yoga groups met the criteria of at least 90 min of running/yoga practice per week. The SD was large because some runners reported running for 400-600 min per week.

How were height and weight measured? Report

Response: Participants’ height and weight were measured without shoes using a portable stadiometer and a digital weighing scale. We have added this information in the text (line 178)

Physiological measures - where and at what time were they done? Who conducted them? Report

Response: All physiological measures were conducted in the laboratory between 8 am and 11 am by two trained experimenters. This information is provided in the text where appropriate (line 137 and 174)

A flowchart of procedures would be beneficial. Please add

Response: We added a flowchart of study procedures as the reviewer suggested to the manuscript as Figure 1.

Step test - was there any demonstration, any warm-up, report the metronome model. How long after the physiological measurements was this done? Report

Response: The experimenter demonstrated the step test procedures and led a 5-minute warm-up before the actual 3-minute step test initiated (line 186). The frequency of the step was guided by an audio file downloaded from the Sports Administration website. Reference and texts were added to the manuscript (line 151). After the physiological measurements, the participants were instructed to rest for 10 minutes before they performed the step test. This information is provided in the text (line 185)

Stressors - what results were recorded and put forward into analysis and where are the descriptive results presented as they are not in table 1? report

Response: In this present study, the purpose of stress stimuli was to induce mental stress and not to serve as a stand-alone measurement tool. Therefore, the correct rate and response time of the stress stimuli for each participant was not reported and we do not believe reporting these data is necessary for interpreting the findings of this present study.

How were your participants divided into 3 groups? report

Response: The participants were assigned to each group according to the types of physical activities that they regularly performed. For example, participants who perform yoga practice (or running) and met the criteria for the present study were assigned to the yoga (or running) group. Participants who did not regularly perform any type of physical activity were assigned to the sedentary group (line 97).

The limitations of the study paragraph could be extended.

Response: We added the statistical power of 0.80 as additional limitation to the manuscript (line 405). We welcome reviewer’s comment if there is anything else needed to be addressed.

Reviewer 3 Report

Comments to the manuscript ID jerph-1766560 entitled:  The psychophysiological profile and cardiac autonomic reactivity in long-term female yoga practitioners: a comparison with runners and sedentary individuals. This is great research about the difference between yoga practitioners and runners and sedentary subjects to analyze cardiac parameters.

Congratulations to the authors. Is a great research well structure and explained. Minor changes are required:

0 Abstract: Well-structured and is adapted to the research.

1 Introduction: Well explained to conduct the aim of the study.

2.- Materials and methods:

Can authors explained how calculate the sample size?

Procedure: Did author take care the post prandial effect in the data collection? Was the data collection realized at the same time in the day in all the subjects?

3.- Results: Very well explained

4.- Discussions: is a great discussion.

5.- Conclusion: Is adapted to the aim of the research.

Author Response

2.- Materials and methods:

Can authors explained how calculate the sample size?

Response: Thank you for the comment. The sample size was calculated using G*Power version 3.1 with alpha of 0.05 and power of 0.80. The a priori analysis yielded a sample size of 18 for each group. We also added the sample size calculation to the manuscript (line 98).

Procedure: Did author take care the post prandial effect in the data collection? Was the data collection realized at the same time in the day in all the subjects?

Response: Thank you for the comments. All participants were instructed to avoid food or drink containing alcohol or caffeine 4 hours prior to the test. The data collection was conducted between 8 and 11 am for each participant to minimize the effects from any meal and/or the circadian rhythm. This information is provided in the text where appropriate. (line 174)

Round 2

Reviewer 2 Report

Dear Authors

Thank you for addressing the majority of my raised questions, The manuscript improved significantly. However, there are still some minor details that need to be added:

- add the model of weight scale and stadiometer

Abut the limitations of the study paragraph - as you mentioned in one of your responses - you have some that reported up to 600 min of running and some 90min. So if you are a little critical you have people with high physical activity and moderate to low. Therefore, this could affect your results and recovery heart rates as 600 min runners would recover faster. This could also be checked in one of your follow-up studies (just a suggestion) :). Please amend the limitations paragraph. Also, they performed different styles of yoga, which have different difficulties and therefore their effect on the study outcome is not known. Perhaps this could also be an interesting factor to research in the future.

Overall, I congratulate the authors and recommend acceptance after minor revision.

Kind regards

Author Response

1. add the model of weight scale and stadiometer

Response: Thanks for the comments. The model of weight scale and stadiometer were added in the manuscript (line 179)

2. About the limitations of the study paragraph - as you mentioned in one of your responses - you have some that reported up to 600 min of running and some 90min. So if you are a little critical you have people with high physical activity and moderate to low. Therefore, this could affect your results and recovery heart rates as 600 min runners would recover faster. This could also be checked in one of your follow-up studies (just a suggestion) :). Please amend the limitations paragraph. Also, they performed different styles of yoga, which have different difficulties and therefore their effect on the study outcome is not known. Perhaps this could also be an interesting factor to research in the future.

Response: Thanks for the comments. We have added these factors to the manuscript (line 406).